# Cryobanking European Mink (*Mustela lutreola*) Mesenchymal Stem Cells and Oocytes

**DOI:** 10.3390/ijms23169319

**Published:** 2022-08-18

**Authors:** Alexandra Calle, Miguel Ángel Ramírez

**Affiliations:** Department of Animal Reproduction, National Institute for Agriculture and Food Research and Technology (INIA), CSIC, 28040 Madrid, Spain

**Keywords:** European mink, ex situ conservation, biobank, mesenchymal stem/stromal cells

## Abstract

The European mink (*Mustela lutreola*) is one of Europe’s most endangered species, and it is on the brink of extinction in the Iberian Peninsula. The species’ precarious situation requires the application of new ex situ conservation methodologies that complement the existing ex situ and in situ conservation measures. Here, we report for the first time the establishment of a biobank for European mink mesenchymal stem cells (emMSC) and oocytes from specimens found dead in the Iberian Peninsula, either free or in captivity. New emMSC lines were isolated from different tissues: bone marrow (emBM-MSC), oral mucosa (emOM-MSc), dermal skin (emDS-MSC), oviduct (emO-MSc), endometrium (emE-MSC), testicular (emT-MSC), and adipose tissue from two different adipose depots: subcutaneous (emSCA-MSC) and ovarian (emOA-MSC). All eight emMSC lines showed plastic adhesion, a detectable expression of characteristic markers of MSCs, and, when cultured under osteogenic and adipogenic conditions, differentiation capacity to these lineages. Additionally, we were able to keep 227 Cumulus-oocyte complexes (COCs) in the biobank, 97 of which are grade I or II. The European mink MSC and oocyte biobank will allow for the conservation of the species’ genetic variability, the application of assisted reproduction techniques, and the development of in vitro models for studying the molecular mechanisms of infectious diseases that threaten the species’ precarious situation.

## 1. Introduction

The European mink (*Mustela lutreola*) is a semiaquatic carnivorous mustelid species native to Europe [1]. The European mink was widespread across continental Europe [2]; however, due to the species’ decline over the last two centuries, it has now disappeared from the majority of countries [3,4], as evidenced by the IUCN’s successive declaration statuses: 1988, Vulnerable (V); 1994, Endangered (E); 2011, Critically Endangered (CR) (https://www.iucnredlist.org/species/14018/45199861, accessed on 3 March 2015). According to recent reports, the European mink is now only widely dispersed throughout Estonia, Belarus, Russia, Romania, and northern Spain [4,5]. Historically, the main causes of species extinction were habitat destruction and population fragmentation, as well as hunting pressure. However, the main threat to European mink populations in recent years has been the invasive alien species American mink (*Neovison vison*), which has escaped or has been deliberately released from fur fabrics [6]. Attacks or aggressive interactions between the two species in the Iberian Peninsula caused and dominated by American mink have been documented in captivity and in the wild [7,8].

The European mink conservation plan, approved in 2003, includes breeding programs, reintroduction into natural habitats, and the development of other ex situ conservation approaches, such as the establishment of a Genome Resource Bank to improve reproductive technologies [9]. A vital tool for preventing the irreparable loss of biodiversity in a species due to the disappearance of its individuals is the creation of biobanks [10]. The European mink, which is currently distributed in small and dispersed populations, is very vulnerable to inbreeding depression, disasters, epidemics, or threats such as the American mink. Therefore, the establishment of a biobank is vital to avoid the extinction of the species and to preserve its genetic heritage.

For the investigation and protection of a species’ biodiversity, every component of a biobank is crucial [10]. Additionally, biobanks provide direct assistance to breeding facilities, increasing their effectiveness by making the best use of housing space and by preserving biological materials [10]. The value of animal cell biobanks containing germ cells or embryos in assisted reproductive technologies cannot be understated. Germplasm banks contain embryos and gametes that have been preserved to address the lack of contact between animals caused by low densities in natural habitats, long distances between artificial habitats in conservation programs, infertility issues caused by low genetic diversity [11], or difficulties associated with reproductive management of wild animals, among other issues. Methods for the collection and cryopreservation of sperm, as well as artificial insemination with either fresh or frozen–thawed material for mustelids, have mostly been developed in domestic and black-footed ferrets [9]. However, when electroejaculation protocols were used on European and American minks that had been captured in the wild, it was observed that the majority of the latter’s males produced semen of relatively good quality, whereas European mink males were less likely to respond positively and, when they did, the quality of their semen was unsatisfactory [9]. Moreover, oocytes are extremely sensitive to freezing procedures, and no successful standardized freeze–thaw protocols have been described in the mammalian oocyte to date [12]. Thanks to the improvements in methods such as cell reprogramming and somatic cell nuclear transfer, the preservation of somatic cells is an alternative and promising tool in conservation. The capacity to obtain viable gametes and subsequently embryos from induced pluripotent stem cells (IPSC) obtained through somatic cell reprogramming illustrates the significance of maintaining somatic cells in species conservation [13] so that somatic cell banks are currently a contrasted method for preserving the biodiversity of threatened species [14]. Even so, there are still limitations in obtaining gametes from IPSC, so somatic cell biobanking should reinforce sample and cell quality as much as possible. In this regard, mesenchymal stem/stromal cells (MSC) are a preferent source of somatic cells for biobanking.

MSC are multipotent adult cells found in most fetal and adult stromal tissues. Because of their immunomodulatory abilities, low immunogenicity, self-renewal, and multilineage differentiation capacity, MSCs are now widely used in cellular therapy, regenerative medicine, and tissue engineering [15,16]. These MSCs’ properties also enhance and facilitate the biobank’s purpose. However, while MSC can be isolated from almost any tissue, there is no practical method for their direct collection, and they have to undergo in vitro expansions and characterization processes known as “biomanufacturing and biobanking” [17].

There are references to the isolation and establishment of MSC from livestock, laboratory animals, or pets, as well as from endangered species such as the snow leopard, black-footed cat, or red panda [18,19,20]. However, no high-quality biobank of European mink MSC or somatic cells has been reported. Indeed, the number of studies on European mink is relatively low, as evidenced by the number of scientific papers on the species [2]. MSC or somatic cell preservation allows one to learn more about the species and contribute to ART application in the wild, breeding centers, or zoos. Furthermore, MSC biobanks enable the optimization of in vitro models for studying the molecular mechanisms of infectious diseases (Aleutian virus, SARS-CoV-2) to improve the conservation of endangered species. By preserving European mink somatic cells and oocytes, the current work aims to create a high-quality source of genetic resources bank to support the conservation of European mink.

## 2. Results

### 2.1. Isolation of European Mink MSC from Different Tissues

Table 1 shows that we were able to successfully isolate 30 primary cultures from a total of 21 European mink specimens (11 females and 10 males) from different geographical areas of the Iberian Peninsula. Unfortunately, due to the advanced state of decomposition in which they were discovered in the field, isolation of live cells from 12 more animals (five females and seven males) was not possible.

We were able to successfully establish and characterize 26 emMSC lines from the bone marrow, oral mucosa, dermal skin tissue, oviduct, endometrium, testicular stromal tissue, and adipose tissue from two different adipose depots: subcutaneous and ovarian.

The primary cultures derived from peripheral blood, pericardial adipose tissue, and abdominal adipose tissue have not yet been fully established or characterized, so they are not described as cell lines.

All of the emMSC from the eight different tissue sources adhered to the plastic surface of culture dishes in primary culture, displaying different morphologies (round, spindle, or elongated shape) (Figure 1, upper panel). After the first cell passage, however, the cells formed homogeneous populations of adherent epithelial-like cells or fibroblast-like cells (Figure 1, lower panels).

In summary, we have preserved cell lines from eight tissues that have been expanded further than nine passages in cell culture and primary cultures from three tissues that have been expanded in vitro up to cell passages 1-5.

### 2.2. Isolation, Sorting, and Cryobanking of Immature Oocytes

In total, 227 Cumulus–oocyte complexes (COCs) (97 grade I and II and 130 grade III and IV using criteria based on the uniformity of ooplasm and cumulus cell complement [21]) have been isolated and vitrified and are cryopreserved in six cryovials (Figure 2).

### 2.3. Immunocytochemical Analysis by Flow Cytometry

Flow cytometry was used to assess some characteristic cell surface markers to further characterize all eight types of emMSCs (Figure 3 and Appendix A). Except for emOA-MSC, all cell lines examined expressed high levels of MSC-specific markers (CD44, CD9, CD29, and CD90). emOA-MSC expressed high levels of CD44, CD9, and CD29 markers but was negative for CD90.

In terms of pluripotency marker expression, POU5F1 was expressed in all emMSCs except emBM-MSC, which did not express any of the pluripotency markers analyzed (POU5F1, SOX2, and STRO1). The SOX2 marker was found exclusively in the three emMSCs associated with female reproductive tissues.

### 2.4. Differentiation Assay

As shown in Figure 4, eight emMSC lines that were representative of the different cellular sources were cultured in osteogenic conditions and displayed remarkable calcium deposits, indicating a high potential for osteogenic differentiation. When cultured under adipogenic conditions, all the emMSC lines examined were also capable of differentiating into adipocytes, resulting in the formation of cytoplasmic lipid droplets that could be seen after Oil red O solution staining. 

## 3. Discussion

Unfortunately, because the European Mink is so close to extinction, it is crucial to develop, establish, monitor, and maintain a biobank for this species. On the one hand, in situ conservation measures are not producing the expected results due to the difficulty of eradicating the exotic American mink in natural European mink habitats. The American mink not only competes for the same ecological niche as the European mink, but it also poses a threat as a vehicle for infectious pathologies [22,23,24]. On the other hand, reproductive management and reproductive methodologies in breeding centers have limitations in obtaining sufficient healthy litters for release into natural habitats [11,25,26] and in ensuring appropriate genetic diversity.

The primary goal of this work is to contribute to E. mink ex situ conservation by establishing the first biobank of somatic cells and oocytes from specimens from the Iberian Peninsula. The samples for this biobank were taken from 21 specimens (11 females and 10 males) from a total of 33 (16 females and 17 males) received animals. Several key factors have contributed to the successful isolation of the primary cultures of various tissues from specimens found dead in the wild or in breeding facilities or zoos: (a) the speed of refrigerated transport of the cadaver to the laboratory; (b) the time elapsed between the death of the animal and its discovery in the case of carcasses of wild specimens, as well as the meteorological conditions; (c) predation on an animal which increases the exposure of damaged tissues to arthropod larvae and microorganisms; (d) in most cases, the received animals had a low body-fat ratio or even a cachectic condition as a result of prolonged periods of malnutrition and/or convalescence or illness before dying in freedom or breeding centers. This poor physical condition also had an impact on the yield of primary cultures isolated, particularly from tissues such as adipose tissue; (e) many of the specimens processed were adults. The amount of MSCs found in tissues decreases with age [27].

Our research showed that bone marrow, oral mucosa, dermal skin tissue, oviduct, endometrium, testicular stromal tissue, and adipose tissue from two different adipose depots—subcutaneous and ovarian—could all be used to isolate and expand European mink MSCs in vitro. Additionally, primary cultures from pericardial adipose tissue, abdominal adipose tissue, and peripheral blood were isolated. Our findings also show that emOM-MSCs, emDS-MSCs, emOA-MSC, emE-MSC, emT-MSC, and emBM-MSC shared similarities in their epithelial morphology, expression of mesenchymal-related markers, and ability to differentiate into osteocytes and adipocytes. Both emSCA-MSCs and emO-MSCs displayed epithelial cell-like morphology, although they showed both characteristic MSC markers and the ability to differentiate into mesodermal lineages. This observation could relate to a mesenchymal to epithelial transition (MET) in these MSC lines, a process in which mesenchymal cells are reprogrammed, eventually losing mesenchymal cell properties while gaining epithelial cell characteristics [28,29].

Passaged cells had a more uniform morphology and developed into colonies as the culture developed. These morphological observations suggest that the isolated cells may contain both mature and progenitor populations, in line with what we have demonstrated in earlier studies on porcine, bovine, and rabbit species [29,30,31,32]. Our results show that except for emOA-MSC, all of the emMSCs that were isolated were positive for CD9, CD29, CD90, and CD44 characteristic mesenchymal markers not previously reported in European mink. emOA-MSC expressed the remaining characteristic mesenchymal cell markers examined (CD9, CD29, and CD44), as well as the pluripotency markers SOX2 and POU5F1, even though it was negative for CD90. The expression of pluripotency markers by MSC has already been reported in other animal species such as cows [33,34] or pigs [32]”. “The gene expression profile of adipose MSC derived from subcutaneous fat in Zucker rats was discovered to be distinct from the gene expression profiles of the other four visceral depots (epicardial, epididymal, mesenteric, and retroperitoneal) [35]. Furthermore, differences in adipose-MSC from mesenteric and omental depots have been observed [36]. Individual visceral “subdepots” should not be considered interchangeable, according to these findings. emOA-MSC could be in a mesenchymal to epithelial transition (MET).” CD29 and CD90 were reported in the literature for MSC isolated from adipose tissue from domestic ferrets (*Mustela putorius*) [37]. The tetraspanin CD9 is found on the cell surface of many different cell types and/or cell-derived exosomes. CD9 has also been shown to be expressed by cancer cells and MSC from different tissues [38,39,40,41]. CD9 has been shown to play an important role in MSC proliferation, migration, and adhesion, and it is regarded as a key protein in MSC cell fate [42]. In addition, all established em-MSCs showed expression of pluripotency-related markers POU5F1, except for emBM-MSC. POU5F1, naturally expressed in embryonic and adult stem cells, has been identified as a key regulator of pluripotent stem cell differentiation and self-renewal [43]. Recent studies have reported the detection of POU5F1 or its transcription factor OCT4 in porcine [32], bovine [29,31], or wild European rabbit MSCs [30]. Although the MSC marker STRO1 has been reported in Ferret Dental Pulp Stem Cells, we were not able to detect its expression in none of the here characterized emMSCs. SOX2, a pluripotency marker expressed in MSCs isolated from various tissues of different species such as red pandas [18,44], giant pandas [45], and wild European rabbits [30], was only observed in emO-MSC, OA-MSC, and emE-MSC. One of the main defining characteristics of a typical mesenchymal pattern is the ability of MSC to differentiate in vitro into mesodermal lineages [29,30,31,32], which was proved here for the European mink MSC isolated from different tissues. A total of six cell lines were obtained from the dermal skin. Because of its therapeutic potential, dermal skin is an interesting source of MSC. Aside from cell biobanking, several studies in veterinary medicine have reported the improvement of chronic wounds by the action of combined treatments with MSCs [46,47], representing a very interesting approach for possible treatments of animals in captivity. Oral mucosal MSC isolation has so far only been reported in humans [48] and rabbits [30]. We have reported here the isolation of five emOM-MSC.

The cryopreservation of sperm and embryos has advanced significantly in germ cell banks; however, oocytes are extremely sensitive to freezing procedures, and no successful standardized freeze–thaw protocols have been described in most species to date [12]. In this study, we employ the COC cryofreezing methodology that has been successfully used on canine species that are phylogenetically close to European minks. We were able to keep 227 COCs in the biobank using this approach, 97 of which are grades I and II. In addition, the size of the COCs isolated and characterized in grades I and II corresponds to the size of *Mustela vison* primordial follicles (approximately 0.5 mm), as reported by Enders, Robert K [49]. Cryopreserved European mink oocytes open the way for the development of assisted reproductive technologies for species conservation, such as in vitro maturation, in vitro fertilization, or intracytoplasmic sperm injection and embryo transfer.

The development of oocyte cryopreservation is critical for the establishment of genetic banks as well as the development of applications for the conservation of endangered European mink. In addition, keeping MSC in biobanks is a good way to ensure the genetic diversity of many endangered species [18,45,50]. This emMSC biobank will also enable the development of in vitro models for toxicologic, epidemiologic, or infectious disease studies [51,52,53], which is a valuable tool for understanding the molecular mechanisms of potential animal health threats that can affect endangered species.

By developing the first biobank of somatic cells and oocytes from specimens from the Iberian Peninsula, we have contributed to the ex situ conservation of the European mink.

## 4. Materials and Methods

### 4.1. Isolation and Culture of European Mink Cell Lines

Multiple different tissue samples were obtained post-mortem from 21 specimens (10 males and 11 females) found dead in the wild, breeding centers, or zoos over the last ten years (2013–2022) as part of the ex situ conservation program of the European mink (*Mustela lutreola*) in Spain.

For the isolation of primary cultures from the oral mucosa, dermal skin tissue, oviduct, endometrium, testis, bone marrow, and adipose tissue from different adipose depots: subcutaneous, ovarian, abdominal, and pericardial, the procedures have been detailed [29,30,32]. Briefly, the collected samples were minced before being incubated in 0.05% collagenase type II, 0.1% BSA, and 30 nM CaCl_2_. Thereafter, a volume of culture medium was added to block the action of collagenase. The resulting pellets were resuspended in DMEM-LG supplemented with 10% FCS, 2 mM glutamine, 1 mM MEM nonessential amino acid solution, and antibiotics (100 U/mL penicillin, 100 mg/mL streptomycin), plated in a 100-mm^2^ tissue culture dish (Jet-Biofil, Guangzhou, China), and incubated in an atmosphere of humidified air and 5% CO_2_ at 37 °C. The culture medium was changed every 48–72 h. For the isolation of the peripheral blood primary cultures, the procedures have been detailed [31]. Briefly, blood samples were diluted 1:1 in phosphate-buffered saline (PBS) and layered onto Biocoll separating solution (Biochrom AG, Germany) that was centrifuged at 1600× *g* for 20 min. The cells were recovered at the interphase, washed, and suspended in a culture medium. Isolated colonies of putative emMSCs were apparent after 5–6 days in culture and were maintained in a growth medium until ~75% confluence. The cells were then treated with 0.05% trypsin–EDTA (T/E) and further cultured for subsequent passage in 100-mm^2^ dishes at 50,000 cells/cm^2^.

### 4.2. Collection and Vitrification of Immature Oocytes

TCM 199 (Biowest, Nuaillé, France), supplemented with 25 mM HEPES, 0.1 mM glutamine, and 1% penicillin–streptomycin, was used to rinse each ovary dissected from the ovarian bursa. Cumulus-oocyte complexes (COCs) were washed and graded using criteria based on ooplasm uniformity and cumulus cell complement, as previously described, under a stereomicroscope (200×) [21]. The grade 1 oocytes had a homogeneous dark cytoplasm and were surrounded by more than five layers of compact cumulus cells. The grade 2 oocytes had a homogeneous dark cytoplasm and were surrounded by three to five layers of compact cumulus cells. The grade 3 oocytes lacked homogeneity in the cytoplasm and were partially surrounded by cumulus cells. The grade 4 oocytes were denuded (lack of cumulus cells), and the cytoplasm was not homogeneous [54]. COCs were cryopreserved according to Turanthum’s vitrification procedure [54].

### 4.3. Immunocytochemical Analysis of emMSC by Flow Cytometry

The cell cultures at 80–90% confluence were detached and fixed with 4% paraformaldehyde for 10 min. The stainings were carried out as described by Calle et al. [31], employing anti-CD44 (clone IM7 Rat, Bio-Rad-MCA806GA. Hercules, CA, USA); Anti-CD9 (clone VJ 1/20, Immunostep, Salamanca, Spain), Anti-CD29/integrinB (clone TS2/16), Anti-CD90/Thy1 and Vimentin (clone LN6 Sigma-Aldrich, Madrid, Spain). Appropriated secondary staining was performed using F(ab’)2-Goat anti-Mouse IgG (H + L) Cross-Adsorbed Secondary Antibody, APC (A10539) from Life Technologies (Carlsbad, CA, USA). To analyze pluripotency, the cells were trypsinized, washed with PBS, and fixed with 4% paraformaldehyde for 10 min. Fixation was neutralized with TBS, and then the cells were incubated with a blocking solution (5% BSA, 0.1% Saponin in PBS) for 30 min at RT. The cells were pelleted and resuspended in the permeabilization solution (15 mM Glycine, 0.1% Saponin, 10 mM HEPES, 0.5% BSA in PBS) and incubated with the primary antibody: Anti-POU5F1 (polyclonal rabbit Biorbyt), anti-SOX2 (clone 20G5, Thermo Fisher Scientific, Meridian Road, Rockford, IL, USA), or anti-STRO1 (clone STRO-1 Sigma-Aldrich) Negative control samples corresponded to the samples in which the primary antibody was omitted. The samples were analyzed in a FACSCanto (BD Biosciences, San Jose, CA, USA) using Flow-JoX Soft-ware^®^ version 10.0.7r2 (TreeStar, Ashland, OR, USA).

### 4.4. In Vitro Differentiation Potential Assay of emMSC

The cells were cultured according to the manufacturer’s instructions for the StemPro^®^ adipogenesis or osteogenesis differentiation kits (Thermo Fisher Scientific, Meridian Road, Rockford, IL, USA) for adipogenic and osteogenic differentiation, respectively, and analyzed as detailed in Calle et al. [29,30,31,32]. Briefly, on a 12-well/24-well multidish, the different emMSC were grown until 90% confluence (JetBiofil, Guangzhou, China). For 14 days, the differentiating media were changed every 2–3 days. After that, the cells were fixed in a 4% paraformaldehyde solution for 10–15 min. After fixation, the cells were incubated in 60% isopropanol for 5 min before being stained with Oil red O (Merck KGaA, Darmstadt, Germany) solution to see the accumulation of red lipid droplets. Photographs of cells were taken with an inverted Nikon Diaphot phase contrast microscope and a Jenoptik ProgRes CT1 digital camera.

The differentiating media were changed every 3–4 days for 21 days to promote osteogenic differentiation. After that, the cells were fixed in a 4% paraformaldehyde solution for 30 min. After fixation, the cells were incubated in a 2% Alizarin Red S solution (pH 4.2) for 2–3 min to visualize the calcium deposits.

## 5. Conclusions

In summary, this study describes the establishment of a biobank for European mink somatic cells and oocytes from specimens found dead in the Iberian Peninsula, either free or in captivity. The cell lines established and characterized from different tissues of origin show a clear mesenchymal pattern. The European mink MSC and oocyte biobank allows for the conservation of the species’ genetic variability, the application of assisted reproduction techniques, and the development of in vitro models for studying the molecular mechanisms of infectious diseases that threaten the species’ already precarious situation.

## Figures and Tables

**Figure 1 ijms-23-09319-f001:**
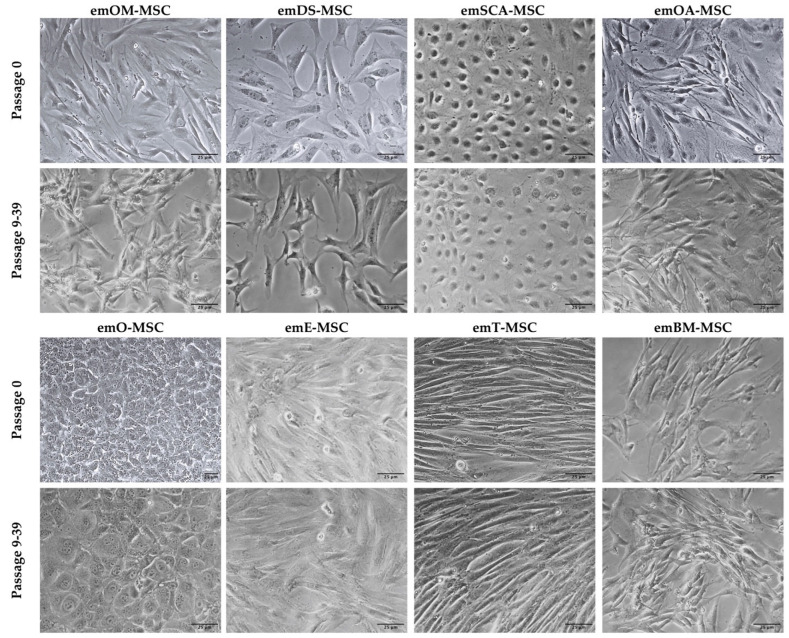
Phase-contrast images of different European mink MSC lines at Passage 0 (upper panels) and Passages 9-39 (lower panels). ×200 magnification. Bars = 25 μm.

**Figure 2 ijms-23-09319-f002:**
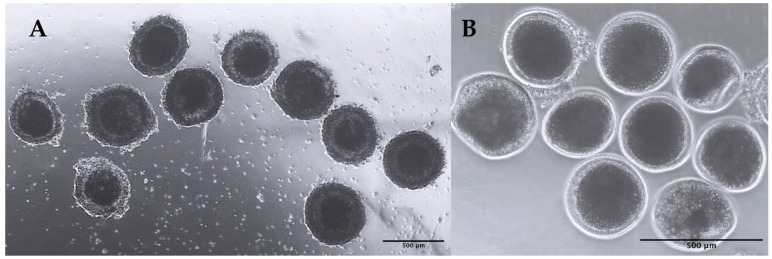
Images of immature European mink oocytes. (**A**) Grade I and II from a specimen found dead on 11 January 2017. (**B**) Grade 4 specimen found dead on 7 July 2016.

**Figure 3 ijms-23-09319-f003:**
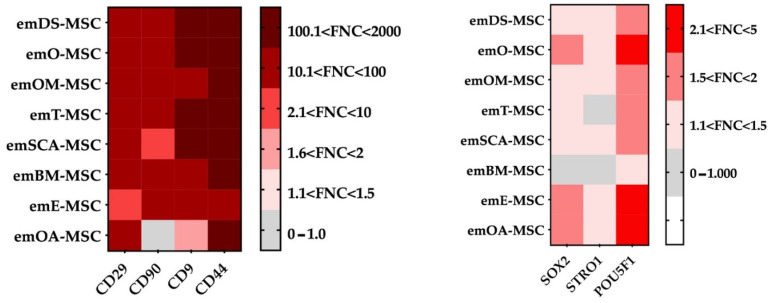
Analysis by flow cytometry of the expression levels of cell surface markers CD29, CD90, CD9, CD44, SOX2, STRO1, and POU5F1 in emDS-MSC, emO-MSC, emOM-MSC, emT-MSC, emSCA-MSC, and emBM-MSC, emE-MSC, and emOA-MSC. Heat map showing the mean fluorescence intensity (folds of negative control (FNC) in the absence of primary antibody) for each sample.

**Figure 4 ijms-23-09319-f004:**
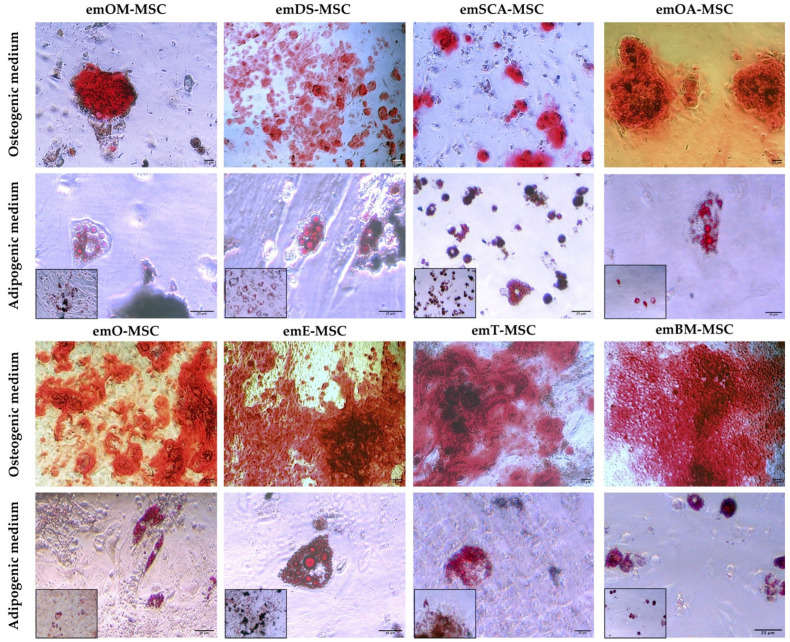
In vitro differentiation of emMSC to different lineages. Images show Alizarin Red S staining of calcium deposits in cells cultured in osteogenic differentiation medium (top panels); and Oil red O staining of lipid droplets in cells cultured in adipogenic differentiation medium (bottom panels). Bright-field images were acquired with 100× magnification (bars = 25 μm) for top panels and with 400× or 200× magnification (bars = 25 μm) for bottom panels.

**Table 1 ijms-23-09319-t001:** European mink primary cultures and cell lines biobank content. Summary of primary cultures and established cell lines preserved in the biobank, organized by source, animal gender, number of passages, and number of cryovials.

Source (European Mink Tissue)	N (Primary Cultures)	N (Cell Lines)	Animal ID	Cell Passages	Nº Vials
Peripheral blood tissue derived cells (emPB-C)	1		89 ♂	5	7
Bone marrow mesenchymal stem cells (emBM-MSC)		2	89 ♂	12	15
			90 ♂	4	4
Abdominal adipose tissue-derived cells (emAA-C)			90 ♀	1	0
	1		79 ♀	1	2
Subcutaneous adipose mesenchymal stem cells (emSCA-MSC)		2	78 ♂	25	34
			89 ♂	6	10
Dermal skin mesenchymal stem cells (emDS-MSC)		6	75 ♀	33	29
			77 ♀	4	2
			78 ♂	12	14
			80 ♂	3	1
			89 ♂	5	3
			90 ♀	1	1
Oral mucose mesenchymal stem cells (emOM-MSC)		5	78 ♂	15	23
			79 ♀	3	1
			80 ♀	6	8
			89 ♂	3	2
			90 ♀	14	10
Pericardial adipose tissue-derived cells (emPA-C)	1		90 ♀	5	5
Oviductal mesenchymal stem cells (emO-MSC)		4	69B♀	11	21
			71 ♀	49	56
			79 ♀	7	7
			90 ♀	6	6
Endometrial mesenchymal stem cells (emE-MSC)		2	79 ♀	5	10
			90 ♀	9	16
Ovarian adipose mesenchymal stem cells (emOA-MSC)		2	69B♀	5	2
			77 ♀	42	53
Testicular mesenchymal stem cells (emT-MSC)		3	69 ♂	45	53
			78 ♂	4	4
			89 ♂	6	6
TOTAL	3	26	11 ♀/10 ♂		405

## Data Availability

The data that support the findings of this study are available from the corresponding author upon reasonable request.

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
