# Peer review of "Cryobanking European Mink (Mustela lutreola) Mesenchymal Stem Cells and Oocytes"

_ijms, 2022, doi:10.3390/ijms23169319_

Round 1

Reviewer 1 Report

Major

Bone marrow (BM) and abdomianl adipose (AA) tissue are typical sources of MSCs but others are not. Especially fibroblasts are a major population of dermal skin and oral mucosa. In addition, adipocytes and preadipocytes are major populations of the subcutaneous fat layer. Others (ovarian adipose, oviduct, endometrium, testis) are very minor sources of MSCs. How can you convince these cells are not other cell types, such as fibroblast and adipocyte. 

Some cells do not express the typical markers (CD90) of MSCs. Differentiation ability is very low. For example, adipogenic differentiation of emOM-MSC, emOA-MSC, and emT-MSCs are only single cells. Therefore, it seems not all cell lines are MSCs.

Minor.

1. Labelings of figures are not insufficient. It seems Figure3a is only 1 sample of FACS. What is the sample name among the 8 cell lines. Although you  showed heat map of each sample, it is better you show other FACS data of 7 cell lines like Figure 3a.

2. Larger view of differentiation pictures is also needed. It is not sufficient to show only a few cells. In addition, quantification of the percentage of differentiated cells is needed.

3. Typically, MSCs test differentiation of mesodermal lineages (Adipogenic, osteogenic and chondrogenic). The authors need to perform the chondrogenic differentiation assay as well.

Author Response

Reviewer 1:

Major

Bone marrow (BM) and abdomianl adipose (AA) tissue are typical sources of MSCs but others are not. Especially fibroblasts are a major population of dermal skin and oral mucosa. In addition, adipocytes and preadipocytes are major populations of the subcutaneous fat layer. Others (ovarian adipose, oviduct, endometrium, testis) are very minor sources of MSCs. How can you convince these cells are not other cell types, such as fibroblast and adipocyte.

  1. The isolation and characterization of MSC from different human tissues (adipose tissue, amniotic fluid, amniotic membrane, dental tissues, endometrium, limb bud, menstrual blood, peripheral blood, placenta, and fetal membrane, salivary gland, skin and foreskin, sub-amniotic umbilical cord lining membrane, synovial fluid, and Wharton's jelly) are widely reported in the literature [1] [2] [3].
  2. MSC from different animal species (pig, cow, rabbit) and different sources (dermal skin tissue, endometrium, peripheral blood, oral mucosa, oviduct, mammary gland, subcutaneous adipose tissue, and visceral adipose depots: abdominal and ovarian adipose tissues) have already been reported by our group [4–9].
  3. With the goal of standardization, the International Society for Cellular Therapy proposed three criteria in 2006 to define the minimal characteristics of MSCs [10]: They should exhibit plastic adherence when maintained in standard culture conditions in tissue culture flasks; more than 95 percent of the MSC population should express mesenchymal specific markers, and they should be able to differentiate to mesodermal cell lineages in vitro under standard differentiating conditions. In this article, we show that all eight emMSC lines fulfilled all the requirements that define them as MSC.
  4. The cell lines referenced by the reviewer (emO-MSCs, emOA-MSCs, and emTMSCs) have already accumulated more than 40 passages in vitro culture. One feature of mesenchymal cells that differentiated cells (fibroblasts, adipocytes) lack are their almost unlimited proliferation capacity [11].
  5. All established cell lines have cell diameters that correspond to those reported in the literature for MSC (10 to 30 μm) [12].

Some cells do not express the typical markers (CD90) of MSCs. Differentiation ability is very low. For example, adipogenic differentiation of emOM-MSC, emOA-MSC, and emT-MSCs are only single cells. Therefore, it seems not all cell lines are MSCs.

To further clarify the reviewer's concerns, a paragraph was rewritten (Line 201) and a new paragraph was added (Line 204).

“Our results show that except for emOA-MSC, all emMSCs isolated were positive for CD9, CD29, CD90, and CD44 characteristic mesenchymal markers not previously reported in European mink.”

“emOA-MSC expressed the remaining characteristic mesenchymal cell markers examined (CD9, CD29, and CD44), as well as the pluripotency markers SOX2 and POU5F1, even though it was negative for CD90. The expression of pluripotency markers by MSC has already been reported in other animal species such as cows [13, 14] or pigs [9]”.

“The gene expression profile of adipose MSC derived from subcutaneous fat in Zucker rats was discovered to be distinct from the gene expression profiles of the other four visceral depots (epicardial, epididymal, mesenteric, and retroperitoneal) [15]. Furthermore, differences in adipose-MSC from mesenteric and omental depots have been observed [16]. Individual visceral "subdepots" should not be considered interchangeable, according to these findings. emOA-MSC could be in a mesenchymal to epithelial transition (MET).”

In response to the reviewer's suggestion, we have made a new figure 4 with new images that are more indicative of the outcomes of the adipogenic differentiation of the different emMSCs.

Minor.

  1. Labelings of figures are not insufficient. It seems Figure3a is only 1 sample of FACS. What is the sample name among the 8 cell lines. Although you showed heat map of each sample, it is better you show other FACS data of 7 cell lines like Figure 3a.

Following the reviewer's suggestion, we now show in a supplementary figure (Figure S1) the complete FACS data corresponding to all the MSCs analyzed and all the antibodies used.

  1. Larger view of differentiation pictures is also needed. It is not sufficient to show only a few cells. In addition, quantification of the percentage of differentiated cells is needed.

To further clarify the reviewer's concerns, we have added a new reduced image, in the lower left corner of each image, demonstrating the high performance in adipocyte differentiation at a 100x magnification.

  1. Typically, MSCs test differentiation of mesodermal lineages (Adipogenic, osteogenic and chondrogenic). The authors need to perform the chondrogenic differentiation assay as well.

We thank the reviewer for giving us the opportunity to clarify that we have indeed also analyzed the different European mink MSCs for their chondrogenic differentiation capacity. We used the StemPro® chondrogenesis differentiation kit (Thermo Fisher Scientific). Our group has reported the efficacy of these differentiation kits in the characterization of MSCs from different animal species: cow, pig, and rabbit [5, 7–9]. But unfortunately, it has not been suitable for the European mink. The differentiation kit caused high cell death in all lines analyzed through the differentiation process in the two experiments performed, so the results were not considered consistent. Below, we show an Image showing Alcian blue staining of emO-MSC, less affected by cell death, cultured in a
chondrogenic differentiation medium.

Figure: In vitro differentiation of emO-MSC to chondrogenic lineage. Images show Alcian blue staining of acidic proteoglycan in cells cultured in a chondrogenic differentiation medium. The bright-field image was acquired with 100× magnification (bars = 25 μm).

The lack of prior studies on European mink MSC in the literature severely impedes and retards its scientific advancement.

References

  1. Ullah, I.; Subbarao, R.B., Rho, G.J. Human mesenchymal stem cells - current trends and future prospective. Biosci Rep. 2015, 35, e00191. https://doi.org/10.1042/BSR20150025
  2. Marquez-Curtis, L.A.; Janowska-Wieczorek, A.; McGann, L.E., Elliott, J.A. Mesenchymal stromal cells derived from various tissues: Biological, clinical and cryopreservation aspects. Cryobiology. 2015, 71, 181-197. https://doi.org/10.1016/j.cryobiol.2015.07.003
  3. Mushahary, D.; Spittler, A.; Kasper, C.; Weber, V., Charwat, V. Isolation, cultivation, and characterization of human mesenchymal stem cells. Cytometry A. 2018, 93, 19-31. https://doi.org/10.1002/cyto.a.23242
  4. Calle, A., Ramírez, M.Á. Mesenchymal Stem Cells in Embryo-Maternal Communication under Healthy Conditions or Viral Infections: Lessons from a Bovine Model. Cells. 2022, 11, 1858. https://doi.org/10.3390/cells11121858
  5. Calle, A.; Zamora-Ceballos, M.; Bárcena, J.; Blanco, E., Ramírez, M.Á. Comparison of Biological Features of Wild European Rabbit Mesenchymal Stem Cells Derived from Different Tissues. International Journal of Molecular Sciences. 2022, 23, 6420. https://doi.org/10.3390/ijms23126420
  6. Calle, A.; Toribio, V.; Yáñez-Mó, M., Ramírez, M.Á. Embryonic Trophectoderm Secretomics Reveals Chemotactic Migration and Intercellular Communication of Endometrial and Circulating MSCs in Embryonic Implantation. International Journal of Molecular Sciences. 2021, 22, 5638. https://doi.org/10.3390/ijms22115638
  7. Calle, A.; Gutiérrez-Reinoso, M.Á.; Re, M.; Blanco, J.; De la Fuente, J.; Monguió-Tortajada, M.; Borràs, F.E.; Yáñez-Mó, M., Ramírez, M.Á. Bovine peripheral blood MSCs chemotax towards inflammation and embryo implantation stimuli. J Cell Physiol. 2021, 236, 1054-1067. https://doi.org/10.1002/jcp.29915
  8. Calle, A.; López-Martín, S.; Monguió-Tortajada, M.; Borràs, F.E.; Yáñez-Mó, M., Ramírez, M.Á. Bovine endometrial MSC: mesenchymal to epithelial transition during luteolysis and tropism to implantation niche for immunomodulation. Stem Cell Res Ther. 2019, 10, 239. https://doi.org/10.1186/s13287-018-1129-1
  9. Calle, A.; Barrajón-Masa, C.; Gómez-Fidalgo, E.; Martín-Lluch, M.; Cruz-Vigo, P.; Sánchez-Sánchez, R., Ramírez, M.Á. Iberian pig mesenchymal stem/stromal cells from dermal skin, abdominal and subcutaneous adipose tissues, and peripheral blood: in vitro characterization and migratory properties in inflammation. Stem Cell Res Ther. 2018, 9, 178. https://doi.org/10.1186/s13287-018-0933-y
  10. Dominici, M.; Le Blanc, K.; Mueller, I.; Slaper-Cortenbach, I.; Marini, F.; Krause, D.; Deans, R.; Keating, A.; Prockop, D., Horwitz, E. Minimal criteria for defining multipotent mesenchymal stromal cells. The International Society for Cellular Therapy position statement. Cytotherapy. 2006, 8, 315-317. https://doi.org/10.1080/14653240600855905
  11. Bharti, D.; Shivakumar, S.B.; Subbarao, R.B., Rho, G.J. Research Advancements in Porcine Derived Mesenchymal Stem Cells. Curr Stem Cell Res Ther. 2016, 11, 78-93. http://www.ncbi.nlm.nih.gov/entrez/query.fcgi?cmd=Retrieve&db=PubMed&dopt=Citation&list_uids=26201864
  12. Musina, R.A.; Bekchanova, E.S., Sukhikh, G.T. Comparison of mesenchymal stem cells obtained from different human tissues. Bull Exp Biol Med. 2005, 139, 504-509. https://doi.org/10.1007/s10517-005-0331-1
  13. Corradetti, B.; Meucci, A.; Bizzaro, D.; Cremonesi, F., Lange Consiglio, A. Mesenchymal stem cells from amnion and amniotic fluid in the bovine. Reproduction. 2013, 145, 391-400. https://doi.org/10.1530/REP-12-0437
  14. Cardoso, T.C.; Ferrari, H.F.; Garcia, A.F.; Novais, J.B.; Silva-Frade, C.; Ferrarezi, M.C.; Andrade, A.L., Gameiro, R. Isolation and characterization of Wharton’s jelly-derived multipotent mesenchymal stromal cells obtained from bovine umbilical cord and maintained in a defined serum-free three-dimensional system. BMC Biotechnol. 2012, 12, 18. https://doi.org/10.1186/1472-6750-12-18
  15. Ferrer-Lorente, R.; Bejar, M.T., Badimon, L. Notch signaling pathway activation in normal and hyperglycemic rats differs in the stem cells of visceral and subcutaneous adipose tissue. Stem Cells Dev. 2014, 23, 3034-3048. https://doi.org/10.1089/scd.2014.0070
  16. Tchkonia, T. et al. Fat depot-specific characteristics are retained in strains derived from single human preadipocytes. Diabetes. 2006, 55, 2571-2578. https://doi.org/10.2337/db06-0540

Reviewer 2 Report

Ms: ijms-1830791

Title: Cryobanking European mink (Mustela lutreola) mesenchymal stem cells and oocytes

 This paper is novel and interesting. However, the design of the experiment and the presentation of relevant research data still need to be rigorously revised and supplemented to improve the reliability and complete integrity of this research paper.

 Major comments:

1.     Please provide the scale bar in the “Figure 2”.

2.     CD9 is expressed by hematopoietic stem cells and is involved in the differentiation of the megakaryocytic, B-lymphoid and myeloid lineages. Although CD9 is also belong to surface marker for oocyte. Why author CD9 is positive marker for mesenchymal stem cells? Please confirm and rule out this data in the Figure 3a and Figure 3b.

3.     Why only check out the osteogenic and adipose differentiation? Please provide more different tissue differentiation markers such as endothelial (CD31), neural (nestin) and oocytes (CD9 or ooloemma) in this study by immunofluorescence staining assay then make it to be more convince for this study.

4.     The adipoic differentiation ability seems not well in the Figure 4, please provide new data then make it to be more convince

5.     Please provide the quantification data in the Figure 4.

6.     Please provide more detail description of the differentiation condition medium as well as culture time period in the “4.4. In vitro differentiation potential assay of emMSC”.

Author Response

Reviewer 2:

This paper is novel and interesting. However, the design of the experiment and the presentation of relevant research data still need to be rigorously revised and supplemented to improve the reliability and complete integrity of this research paper.

Major comments:

  1. Please provide the scale bar in the “Figure 2”.

Thanks to the reviewer for pointing out our oversight.

Following the reviewer's suggestion, the scale bar was provided in figure 2. Moreover, a paragraph was added in chapter 3 (line 244).

“In addition, the size of the COCs isolated and characterized in grades I and II corresponds to the size of Mustela vison primordial follicles (approximately 0.5 mm), as reported by Enders, Robert K [1].”

  1. CD9 is expressed by hematopoietic stem cells and is involved in the differentiation of the megakaryocytic, B-lymphoid and myeloid lineages. Although CD9 is also belong to surface marker for oocyte. Why author CD9 is positive marker for mesenchymal stem cells? Please confirm and rule out this data in the Figure 3a and Figure 3b.

We thank the reviewer for this comment, which has allowed us to clarify the key role that CD9 plays in MSCs. A paragraph was added in chapter 3 (line 215).

“The tetraspanin CD9 is found on the cell surface of many different cell types and/or cell-derived exosomes. CD9 has also been shown to be expressed by cancer cells and MSC from different tissues [2–5]. CD9 has been shown to play an important role in MSC proliferation, migration, and adhesion, and it is regarded as a key protein in MSC cell fate [6].”

  1. Why only check out the osteogenic and adipose differentiation? Please provide more different tissue differentiation markers such as endothelial (CD31), neural (nestin) and oocytes (CD9 or ooloemma) in this study by immunofluorescence staining assay then make it to be more convince for this study.

We thank the reviewer for giving us the opportunity to clarify that we have indeed also analyzed the different European mink MSCs for their chondrogenic differentiation capacity. We used the StemPro® chondrogenesis differentiation kit (Thermo Fisher Scientific). Our group has reported the efficacy of these differentiation kits in the characterization of MSCs from different animal species: cow, pig, and rabbit [7–10]. But unfortunately, it has not been suitable for the European mink. The differentiation kit caused high cell death in all lines analyzed through the differentiation process in the two experiments performed, so the results were not considered consistent. Below, we show an Image showing Alcian blue staining of emO-MSC, less affected by cell death, cultured in a chondrogenic differentiation medium.

Figure: In vitro differentiation of emO-MSC to chondrogenic lineage. Images show Alcian blue staining of acidic proteoglycan in cells cultured in a chondrogenic differentiation medium. The bright-field image was acquired with 100× magnification (bars = 25 μm)

We thank the reviewer for his proposal to analyze different markers of differentiation, but the lack of prior studies on European mink MSC in the literature severely impedes and slows scientific progress. To carry out this research, various antibodies previously reported for other species were tested. The antibodies used in many cases did not recognize the mink protein. The negative results for the mesenchymal and pluripotency markers CD105, MHCII, and NANOG are shown below.

  1. The adipoic differentiation ability seems not well in the Figure 4, please provide new data then make it to be more convince.

In response to the reviewer's suggestion, we have chosen new images that are more indicative of the outcomes of the adipogenic differentiation of the different emMSCs.

  1. Please provide the quantification data in the Figure 4.

To further clarify the reviewer's concerns, we have added a new reduced image, in the lower left corner of each image, demonstrating the high performance in adipocyte differentiation at a 100x magnification.

  1. Please provide more detail description of the differentiation condition medium as well as culture time period in the “4.4. In vitro differentiation potential assay of emMSC”.

Following the reviewer's suggestion, a paragraph was added in chapter 4.4 (line 317).

“Briefly, on a 12-well/24-well multidish, the different emMSC were grown until 90% confluence (JetBiofil, Guangzhou, China). For 14 days, differentiating media were changed every 2-3 days. After that, the cells were fixed in a 4% paraformaldehyde solution for 10-15 minutes. After fixation, cells were incubated in 60% isopropanol for 5 minutes before being stained with Oil red O (Merck KGaA, Darmstadt, Germany) solution to see the accumulation of red lipid droplets. Photographs of cells were taken with an inverted Nikon Diaphot phase contrast microscope and a Jenoptik ProgRes CT1 digital camera.

Differentiating media were changed every 3-4 days for 21 days to promote osteogenic differentiation. After that, the cells were fixed in a 4% paraformaldehyde solution for 30 minutes. After fixation, the cells were incubated in a 2% Alizarin Red S solution (pH 4.2) for 2-3 minutes to visualize the calcium deposits.”

-Moderate English changes are required.

The entire document has been verified for grammatical and linguistic errors.

References

  1. Enders, R.K. Reproduction in the Mink (Mustela Vison). Proceedings of the American Philosophical Society. 1952, 96, 691-755. http://www.jstor.org/stable/10.2307/3143637
  2. Hsieh, C.C.; Hsu, S.C.; Yao, M., Huang, D.M. CD9 Upregulation-Decreased CCL21 Secretion in Mesenchymal Stem Cells Reduces Cancer Cell Migration. Int J Mol Sci. 2021, 22, 1738. https://doi.org/10.3390/ijms22041738
  3. Halfon, S.; Abramov, N.; Grinblat, B., Ginis, I. Markers distinguishing mesenchymal stem cells from fibroblasts are downregulated with passaging. Stem Cells Dev. 2011, 20, 53-66. https://doi.org/10.1089/scd.2010.0040
  4. Kim, Y.J.; Yu, J.M.; Joo, H.J.; Kim, H.K.; Cho, H.H.; Bae, Y.C., Jung, J.S. Role of CD9 in proliferation and proangiogenic action of human adipose-derived mesenchymal stem cells. Pflugers Arch. 2007, 455, 283-296. https://doi.org/10.1007/s00424-007-0285-4
  5. Gronthos, S.; Franklin, D.M.; Leddy, H.A.; Robey, P.G.; Storms, R.W., Gimble, J.M. Surface protein characterization of human adipose tissue-derived stromal cells. J Cell Physiol. 2001, 189, 54-63. https://doi.org/10.1002/jcp.1138
  6. Holley, R.J.; Tai, G.; Williamson, A.J.; Taylor, S.; Cain, S.A.; Richardson, S.M.; Merry, C.L.; Whetton, A.D.; Kielty, C.M., Canfield, A.E. Comparative quantification of the surfaceome of human multipotent mesenchymal progenitor cells. Stem Cell Reports. 2015, 4, 473-488. https://doi.org/10.1016/j.stemcr.2015.01.007
  7. Calle, A.; Zamora-Ceballos, M.; Bárcena, J.; Blanco, E., Ramírez, M.Á. Comparison of Biological Features of Wild European Rabbit Mesenchymal Stem Cells Derived from Different Tissues. International Journal of Molecular Sciences. 2022, 23, 6420. https://doi.org/10.3390/ijms23126420
  8. Calle, A.; Gutiérrez-Reinoso, M.Á.; Re, M.; Blanco, J.; De la Fuente, J.; Monguió-Tortajada, M.; Borràs, F.E.; Yáñez-Mó, M., Ramírez, M.Á. Bovine peripheral blood MSCs chemotax towards inflammation and embryo implantation stimuli. J Cell Physiol. 2021, 236, 1054-1067. https://doi.org/10.1002/jcp.29915
  9. Calle, A.; López-Martín, S.; Monguió-Tortajada, M.; Borràs, F.E.; Yáñez-Mó, M., Ramírez, M.Á. Bovine endometrial MSC: mesenchymal to epithelial transition during luteolysis and tropism to implantation niche for immunomodulation. Stem Cell Res Ther. 2019, 10, 239. https://doi.org/10.1186/s13287-018-1129-1
  10. Calle, A.; Barrajón-Masa, C.; Gómez-Fidalgo, E.; Martín-Lluch, M.; Cruz-Vigo, P.; Sánchez-Sánchez, R., Ramírez, M.Á. Iberian pig mesenchymal stem/stromal cells from dermal skin, abdominal and subcutaneous adipose tissues, and peripheral blood: in vitro characterization and migratory properties in inflammation. Stem Cell Res Ther. 2018, 9, 178. https://doi.org/10.1186/s13287-018-0933-y

Round 2

Reviewer 1 Report

The authors addressed my previous concerns sincerely.

Author Response

We appreciate the reviewer's constructive criticism, which enabled us to improve the manuscript.

Reviewer 2 Report

The paper has been improved according to the referee's instructions, however, works have not been properly and confidently justified and need major revision before publication.

 1.     Please provide more different tissue differentiation markers such as endothelial (CD31), neural (nestin) and oocytes (CD9 or ooloemma) in this study by immunofluorescence staining assay because it’s very important issue and necessary for this study. Please carry out for these assay then make it to be more convince for this study

  1. Please provide the quantification data in the Figure 4.

Author Response

Reviewer 2:

The paper has been improved according to the referee's instructions, however, works have not been properly and confidently justified and need major revision before publication.

  1. Please provide more different tissue differentiation markers such as endothelial (CD31), neural (nestin) and oocytes (CD9 or ooloemma) in this study by immunofluorescence staining assay because it’s very important issue and necessary for this study. Please carry out for these assay then make it to be more convince for this study.

In our first round of responses to the reviewer, we explained that to conduct this research, we tested several other important MSC antibodies, previously reported for other species, but they did not recognize the mink protein (CD105, MHCII, and NANOG). These results were not shown in the paper due to the non-specificity of commercial antibodies for the European mink species, but we did attach the negative flow cytometry results to the reviewer to attest that the characterization of emMSCs had been broader. The lack of prior research on European mink MSC in the literature significantly impedes and slows scientific progress.

  1. Concerning the reviewer's requested markers, numerous citations in the literature demonstrate that CD31 is not expressed in the MSC of many species tested, including cows [1], pigs [2], rabbits [3], red pandas [4, 5], giant pandas [6], and humans [7]. Furthermore, as our group has shown in bovine species, CD31 is expressed in endothelial cells but not fibroblast-type cells [8].
  2. Concerning the reviewer's request for an analysis of Nestin expression by different emMSCs, Nestin is a marker of neurogenic lineage to investigate the ectodermal differentiation potential. Because MSCs are mesodermal in origin, we believe it is more important to investigate the differentiation capacity of different emMSCs towards other mesodermal lineages, as in the case of chondrocyte differentiation, instead of analyzing the capacity of trans-differentiation into the ectodermal lineage [9]. We have analyzed the different European mink MSCs for their differentiation capacity towards chondrogenic lineage, which has been widely reported for MSC in different species, as we have already responded to the reviewer in the first round. In the two experiments performed, the differentiation kit caused high cell death in all lines analyzed during the differentiation process, so the results were not considered consistent. In the future, we intend to investigate the differentiation capacity of emMSCs toward chondrocyte lineages by developing a species-appropriate in vitro differentiation medium.
  3. Concerning the reviewer's request for an analysis of CD9 expression, all cell lines examined expressed high levels of MSC-specific markers CD9 (Line 136 and Figure 3).
  1. Please provide the quantification data in the Figure 4.

We modified Figure 4 and attached new images more indicative of the outcomes of the differentiation of the different emMSCs lines in our first round of responses to the reviewer. We also include images that demonstrate the effectiveness of differentiation at lower magnifications.

Considering that: i) the commercial differentiation kits are qualitative and the representative images shown in the manufacturer's instructions do not show differentiation rates of 100%; ii) there are no quantitative analyses of MSC cell differentiation based on analysis of cell differentiation images in the literature; and iii) the differentiation capacity of all emMSC lines is greater than 75% for osteocytic and 50% for adipocytic cell lineages, we believe that all emMSC lines fulfill the in vitro differentiation criteria for mesodermal lineages under standard differentiating conditions established by The International Society for Cellular Therapy [10].

  1. Lee, J. et al. Bovine tongue epithelium-derived cells: A new source of bovine mesenchymal stem cells. Biosci Rep. 2020, 40, BSR20181829. https://doi.org/10.1042/BSR20181829
  2. Dariolli, R.; Bassaneze, V.; Nakamuta, J.S.; Omae, S.V.; Campos, L.C., Krieger, J.E. Porcine adipose tissue-derived mesenchymal stem cells retain their proliferative characteristics, senescence, karyotype and plasticity after long-term cryopreservation. PLoS One. 2013, 8, e67939. https://doi.org/10.1371/journal.pone.0067939
  3. Lee, T.C.; Lee, T.H.; Huang, Y.H.; Chang, N.K.; Lin, Y.J.; Chien, P.W.; Yang, W.H., Lin, M.H. Comparison of surface markers between human and rabbit mesenchymal stem cells. PLoS One. 2014, 9, e111390. https://doi.org/10.1371/journal.pone.0111390
  4. Wang, D.H.; Wu, X.M.; Chen, J.S.; Cai, Z.G.; An, J.H.; Zhang, M.Y.; Li, Y.; Li, F.P.; Hou, R., Liu, Y.L. Isolation and characterization mesenchymal stem cells from red panda (Ailurus fulgens styani) endometrium. Conserv Physiol. 2022, 10, coac004. https://doi.org/10.1093/conphys/coac004
  5. An, J.H.; Li, F.P.; He, P.; Chen, J.S.; Cai, Z.G.; Liu, S.R.; Yue, C.J.; Liu, Y.L., Hou, R. Characteristics of Mesenchymal Stem Cells Isolated from the Bone Marrow of Red Pandas. Zoology (Jena). 2020, 140, 125775. https://doi.org/10.1016/j.zool.2020.125775
  6. Liu, Y. et al. Isolation and characterization of mesenchymal stem cells from umbilical cord of giant panda. Tissue Cell. 2021, 71, 101518. https://doi.org/10.1016/j.tice.2021.101518
  7. Musina, R.A.; Bekchanova, E.S., Sukhikh, G.T. Comparison of mesenchymal stem cells obtained from different human tissues. Bull Exp Biol Med. 2005, 139, 504-509. https://doi.org/10.1007/s10517-005-0331-1
  8. Jiménez-Meléndez, A.; Fernández-Álvarez, M.; Calle, A.; Ramírez, M.Á.; Diezma-Díaz, C.; Vázquez-Arbaizar, P.; Ortega-Mora, L.M., Álvarez-García, G. Lytic cycle of Besnoitia besnoiti tachyzoites displays similar features in primary bovine endothelial cells and fibroblasts. Parasit Vectors. 2019, 12, 517. https://doi.org/10.1186/s13071-019-3777-0
  9. Ullah, I.; Subbarao, R.B., Rho, G.J. Human mesenchymal stem cells - current trends and future prospective. Biosci Rep. 2015, 35, e00191. https://doi.org/10.1042/BSR20150025
  10. Dominici, M.; Le Blanc, K.; Mueller, I.; Slaper-Cortenbach, I.; Marini, F.; Krause, D.; Deans, R.; Keating, A.; Prockop, D., Horwitz, E. Minimal criteria for defining multipotent mesenchymal stromal cells. The International Society for Cellular Therapy position statement. Cytotherapy. 2006, 8, 315-317. https://doi.org/10.1080/14653240600855905

Round 3

Reviewer 2 Report

Authors have not been properly responded for this major revision because it's necessary to carry out for question 1 in line with quality for this journal. Therefore, I recommended rejection and re-submission till completed for this data. 

Author Response

Reviewer 2:

Review Report (Round 3)

Authors have not been properly responded for this major revision because it's necessary to carry out for question 1 in line with quality for this journal. Therefore, I recommended rejection and re-submission till completed for this data.

After completing two rounds of review of our article, during which we addressed every single comment of both reviewers in our point-by-point replies. We are firmly convinced that our data complies with the high-quality standards of the journal, as agreed by reviewer 1, but we reached a dead-end with reviewer 2 because the experiments he is demanding are impossible to be carried out.

We want to emphasize that we are more than willing to analyze additional relevant markers in our cells, provided appropriate reagents specific to European mink were available for these analyses. We have done a thorough search on antibody suppliers and we found no suited reagents available specific for Mustela lutreola. Mustela lutreola is phylogenetically distant from the American mink (Nevison vison) and the ferret (Mustela putorius furos), both of which have antibodies reported in the literature. As we detailed in previous replies, we have tested several different antibodies to analyze crossreactivity with European mink and the results are summarized in the current manuscript version in which we have been able to assess the expression of 4 lineage markers, 2 pluripotency markers as well as the differentiation capacity to two out of the three possible different mesodermal lineages. In sum, we think that the characterization of our MSC lines conveys the scientific standards of MSC studies and with the quality standards of the journal, taking into account the challenge that poses to analyzing an exotic species with scarcely suited reagents available.